# Assessment of Lipid Peroxidation Products in Adult Formulas: GC-MS Determination of Carbonyl and Volatile Compounds Under Different Storage Conditions

**DOI:** 10.3390/foods13233752

**Published:** 2024-11-23

**Authors:** Jorge Antonio Custodio-Mendoza, Alexandra Rangel Silva, Marcin A. Kurek, Paulo Joaquim Almeida, João Rodrigo Santos, José António Rodrigues, Antonia María Carro

**Affiliations:** 1Department of Technique and Food Development, Institute of Human Nutrition Sciences, Warsaw University of Life Sciences (WULS-SGGW), 02-776 Warszawa, Poland; marcin_kurek@sggw.edu.pl; 2Laboratório Associado para a Química Verde (REQUIMTE/LAQV)—Departamento de Química e Bioquímica, Faculdade de Ciências, Universidade do Porto, 4169-007 Porto, Portugal; up201504466@edu.fe.up.pt (A.R.S.); pjalmeid@fc.up.pt (P.J.A.); jrodrigosantos@fc.up.pt (J.R.S.); jarodrig@fc.up.pt (J.A.R.); 3Department of Analytical Chemistry, Nutrition and Food Science, University of Santiago de Compostela, 15782 Santiago de Compostela, Spain; 4Health Research Institute of Santiago de Compostela (IDIS), University of Santiago de Compostela, 15782 Santiago de Compostela, Spain; 5Instituto de Materiais (iMATUS), University of Santiago de Compostela, 15782 Santiago de Compostela, Spain

**Keywords:** dispersive liquid–liquid microextraction, enteral nutrition formula, food safety, solid-phase microextraction, risk of exposure, stability analysis, volatile organic compounds

## Abstract

The occurrence of carbonyl compounds and volatile organic compounds (VOCs) in adult formulas is a critical issue in product safety and quality. This research manuscript reports the determination of targeted and untargeted carbonyl compounds and VOCs in adult formulas stored at different temperatures (room temperature, 4 °C, and 60 °C) over one month. Gas chromatography-mass spectrometry was utilized for the sample analysis. Ultrasound-assisted dispersive liquid–liquid microextraction at 60 °C for 20 min facilitated the extraction of six carbonyl compounds, while headspace solid-phase microextraction (HS-SPME) was employed for the determination of untargeted VOCs using a DVB/CAR/PDMS fiber, involving 15 min of equilibration and 45 min of extraction at 40 °C with magnetic stirring. Analytical features of the methods were assessed according to Food and Drug Administration guidelines, and good limits of detection and quantitation, linearity, accuracy, and precision were achieved. Notably, the highest levels of carbonyl compounds were found in high-protein formulas, with quantifiable levels of malondialdehyde, acrolein, and formaldehyde detected and quantified in 80% of samples. Additionally, significant levels of VOCs such as hexanal and 2-heptanone were found in samples stored at elevated temperatures. These findings suggest the importance of protein content and storage conditions in the levels of carbonyl compounds and VOCs found in adult formulas, with implications for consumer safety and quality control.

## 1. Introduction

Adult formulas are specially designed nutritional products for enteral feeding, providing essential nutrients to individuals who may have difficulty consuming regular food [1,2]. These formulas play a vital role in the nutrition and health of various adult populations, particularly those with chronic illnesses, disabilities, or other health problems requiring specialized dietary support [2,3]. For instance, patients recovering from surgery, individuals with malabsorption issues, and the elderly often rely on these products to meet their nutritional needs and improve their overall health outcomes [1,2,3]. Stringent regulatory standards govern the manufacturing of adult formulas to ensure safety and efficacy [4]. Formulations can vary widely, including standard, high-protein, and specialized options tailored for specific health conditions [3,4]. Ultra-high temperature treatment, commonly used in adult formula processing, rapidly heats products to 135–150 °C for a few seconds to destroy harmful microorganisms and extend shelf life without refrigeration [5]. However, this process can also lead to the formation of heat-induced contaminants, such as lipid peroxidation products and advanced glycation end-products (AGEs), which may pose health risks to consumers [6].

Among the most concerning contaminants are malondialdehyde (MDA), formaldehyde (FCHO), acetaldehyde (ACE), acrolein (ACRL), methylglyoxal (MGO), diacetyl (DA) (the structural formula is provided in the Appendix A), and other various volatile organic compounds (VOCs) [7]. These compounds are formed during lipid peroxidation as polyunsaturated fatty acids interact with reactive oxygen species [6,7]. This process follows a radical-driven pathway, forming lipid hydroperoxides as primary oxidation products [7]. These hydroperoxides decompose into more stable secondary products, including carbonyl compounds such as malondialdehyde, formaldehyde, acetaldehyde, acrolein, methylglyoxal, and diacetyl [7]. These compounds indicate oxidative degradation in lipid-containing foods like adult formula [6,7]. Lipid peroxidation can continue when the formulas are stored along with other degradation processes, potentially affecting their nutritional quality and safety [6,7]. Monitoring the presence and concentrations of these contaminants is crucial due to their toxicological profiles and classifications by health authorities, including the International Agency for Research on Cancer (IARC), which identifies several of these compounds as potential carcinogens [7]. For instance, FCHO is classified as a carcinogen with sufficient evidence of its carcinogenicity to humans (Group 1) with a tolerable daily intake (TDI) set at 150 µg/kg body weight (bw)/day. ACRL is classified as probably carcinogenic to humans (Group 2A), with a lower TDI of 7.5 µg/kg bw/day. ACE is categorized as possibly carcinogenic to humans (Group 2B), suggesting it may be carcinogenic, with an Acceptable Daily Intake (ADI) of 185 µg/day. MDA falls into the group of substances with insufficient evidence for their carcinogenicity (Group 3). However, it possesses a Threshold of Toxicological Concern set by the International Programme on Chemical Safety (IPCS) of 30 µg/kg bw/day. MGO is also in Group 3 and does not have a specific TDI. Finally, while unclassified by the IARC, DA has a notable ADI of 900 µg/kg bw/day [7,8,9,10,11,12].

Given the potential health implications of these contaminants, the need for reliable analytical methods to assess their levels in adult formulas is paramount [7]. Traditional methods for determining lipid peroxidation products in foods, such as the thiobarbituric acid (TBA) test and peroxide value measurement, are widely used due to their simplicity and accessibility [7,13,14]. However, these methods have several limitations: they are non-specific (which can lead to overestimation), time-consuming, and require harsh conditions and large amounts of solvents, which have a notable environmental impact [13,14]. To address these limitations, chromatographic methods such as high-performance liquid chromatography (HPLC) and gas chromatography (GC), paired with various detection systems (UV, Fluorescence Detection (FLD), Flame Ionization Detection (FID), and Mass Spectrometry (MS)), have been developed [7,13,14,15,16,17,18,19,20]. These methods allow for the specific determination of lipid peroxidation products, including aldehydes, ketones, malondialdehyde, and dicarbonyl compounds, with higher sensitivity and selectivity than traditional methods [7]. However, to detect specific compounds like AGEs and other lipid peroxidation products effectively, chemical derivatization (e.g., with hydrazine, TBA, or o-phenylene diamine) is often required to enhance instrument response, depending on the detector system [13,17,18,19,20]. Microextraction techniques have further advanced the sensitivity and specificity of carbonyl compounds and VOCs analysis by offering a more sustainable approach due to the reduced sample size and waste generation and the reduction (Dispersive Liquid–Liquid Microextraction, DLLME) or non-use (Solid Phase Microextraction, SPME) of organic solvent [7]. In this context, head-space (HS) SPME has been used to determine carbonyl compounds associated with lipid peroxidation in infant formula, milk, and dairy products [7,20,21]. VOC determination without derivatization has also been reported using HS-SPME [7]. DLLME has been applied to measure FCHO, MDA, and ACRL [7,16,22] in beverages. To our knowledge, there is no analytical method for assessing lipid peroxidation products in adult formula products.

In this study, we aim to apply innovative microextraction techniques to determine the concentrations of carbonyl compounds and to monitor VOCs in adult formulas. Specifically, we propose using DLLME for the targeted analysis of carbonyl and dicarbonyl compounds as potential markers of lipid peroxidation and HS-SPME for untargeted VOC determination, as it requires no derivatization. Through these methods, we present the first known data on the occurrence of carbonyl compounds associated with oxidative and storage stability in adult formulas, as far as we know. Additionally, we assess the potential risk of exposure for consumers to these contaminants, highlighting the need for ongoing monitoring to warrant the safety and quality of adult formula.

## 2. Materials and Methods

### 2.1. Chemicals and Reagents

Unless otherwise specified, all chemicals used were high purity (>98%). Acetaldehyde (ACE, CAS 75-07-0), Acetonitrile (ACN, CAS 75-05-8), Chloroform (CHCl_3_, 35%, CAS 67-66-3), Deuterated Acetaldehyde (ACEd4, CAS 1632-89-9), Deuterated Acetone (ACOd6, CAS 666-52-4), Diacetyl (DA, CAS 431-03-8), 2,4-Dinitrophenylhydrazine (DNPH, CAS 119-26-6), Formaldehyde (FCHO, CAS 50-00-0), Malondialdehyde (MDA, CAS 643-12-9), and Methylglyoxal (MGO, 40% CAS 78-98-8) were all supplied by Merck (Darmstadt, Germany). Acrolein (ACRL, CAS 107-02-8) was purchased from LGC Standards (London, UK).

SPME fibers with 50/30 μm divinylbenzene/Carboxen/Polydimethylsiloxane (DVB/CAR/PDMS) and 65 μm PDMS/DVB were also obtained from Merck. All fibers were conditioned to ensure optimal performance per the manufacturer’s instructions before their initial use. Manual sampling was performed using a manual holder sourced from Merck. This study utilized various pieces of laboratory equipment, including a Centromix II-BL Centrifuge from J. P. Selecta (Barcelona, Spain), a Basic 20 pH meter from Crison Instruments (Barcelona, Spain), a 2510EMTH ultrasonic bath from Branson Ultrasonics (Danbury, CT, USA), and a Reax Top vortex mixer from Instruments GmbH & Co. (Schwalbach, Germany) to conduct all experiments.

### 2.2. Sample Selection and Stability Study Design

In this study, 12 commercially available adult nutritional formulas were selected from supermarkets and parapharmacies in Santiago de Compostela, Spain. The formulas were categorized into three groups: high-protein formulations (HP, *n* = 5), standard formulations (SAF, *n* = 4), and specialized formulations (SF, *n* = 3) encompassing an enteral formula for dysphagia and amylase resistance, a carbohydrate module formula, and an enteral powder for administration via tube or orally. All samples were kept in their original packaging before the occurrence study. From these, two unflavored standards and two unflavored high-protein adult formulas with similar compositions were selected for the storage stability study.

The stability study aimed to evaluate the effects of different storage temperatures on the formulations over 30 days. Unflavored samples, including high-protein (*n* = 2) and standard formulation (*n* = 2), were selected for this study. All samples were transferred into amber vials and the caps were covered with parafilm to prevent contamination. The formulas were stored in their original powdered form and were prepared according to the manufacturer’s instructions by dissolving them in drinking water before analysis. Three storage conditions were tested: room temperature, 4 °C, and 60 °C. These conditions were selected to replicate typical domestic storage situations throughout the year. The samples were stored in the dark, using amber vials covered with aluminum foil and placed inside cardboard boxes to minimize light exposure. The four samples were analyzed on specified days (0, 1, 7, 14, 21, and 30). In the untargeted VOC determination, focusing on determining VOC changes during storage, two unflavored samples (also used in the target analysis) from the standard formulation category were selected for HS-SPME-GC-MS analysis after 30 days of storage, as they were the only formulations available in individual packaging of small portions. All analyses were conducted in triplicate to ensure reliable data.

### 2.3. Ultrasound-Assisted Dispersive Liquid–Liquid Microextraction

To extract FCHO, MDA, ACE, ACRL, MGO, and DA from the adult formula, a previously developed method to extract MDA, ACRL, and 4-hydroxy-2-nonenal in beverages was used (Custodio-Mendoza et al., 2022) with modifications [22]. The adult formula was prepared following the manufacturer’s instructions: 15% *w*/*v* in drinking water at 40 °C, with gentle stirring until fully dissolved. A 0.5 mL aliquot of the solution was transferred to a 2.5 mL falcon tube, and 1.3 mL of ACN was added. The mixture was vortexed for 1 min to combine the phases, followed by centrifugation for 2 min at 1634 g to precipitate proteins. The supernatant was transferred to a clean tube, and 90 µL of CHCl3 was added. The mixture was homogenized by three cycles of charging and discharging using a Pasteur pipette, then rapidly transferred to a conical tube containing 4 mL of ultrapure water and 0.5 mL of DNPH solution (0.5 g/L in 2 M HCl). The system was incubated in an ultrasonic water bath at 60 °C for 20 min, followed by centrifugation for 2 min at 2146× *g* to separate the extractant phase. The drop containing the carbonyl-DNPH derivatives was collected with a microsyringe and directly injected into the GC-MS for target analysis.

### 2.4. Head-Space Solid Phase Microextraction

The VOCs were extracted from the adult formula following the method described by Clarke et al., (2019) with modifications [21]. Briefly, the adult formula was prepared as previously outlined. A 2 mL aliquot of the sample solution was placed in a 6 mL vial with a magnetic stirrer and sealed with a septum cap. The sample was equilibrated in a water bath with gentle stirring for 15 min at 40 °C. After the equilibration period, the SPME fiber was exposed, and extraction started for 45 min at the same temperature. Following extraction, the analytes were desorbed at 270 °C for 5 min in the GC-MS injection port.

### 2.5. Gas Chromatography-Mass Spectrometry

The targeted and untargeted analyses were performed using a GC–MS system (7890B-5977B, Agilent Technologies, Santa Clara, CA, USA) with a J&W HP-5MS column (30 m × 0.25 mm ID × 0.25 µm, Agilent) and a carrier gas flow rate of 1.5 mL/min. The transfer line was set to 280 °C, connecting the column to an electron impact (EI^+^) source at 250 °C with 70 eV, and a single quadrupole mass analyzer was maintained at 120 °C.

For the targeted analysis, the injector temperature was set to 245 °C using an ultra-inert double taper liner in splitless mode. The temperature program began at 100 °C, increased at 100 °C/min to 230 °C (held for 3.3 min), then ramped at 35 °C/min to a final temperature of 280 °C, maintained for 2 min. The total analysis time for this targeted method was 8 min. Initially, full scans were performed over a 50–300 *m*/*z* range for each hydrazone, allowing the identification of the most intense ions. Single ion monitoring mode was then used with the quantifier and qualifier ions summarized in Appendix A.

For the untargeted analysis, the oven temperature started at 35 °C, held for 5 min, and increased at 5 °C/min to 280 °C (held for 10 min), resulting in a total run time of 64 min. The detector was turned off after 60 min. The injector was set to 270 °C using a straight ultra-inert liner (5190-4048) with a 0.755 mm inner diameter in splitless mode. Scan acquisition mode covered a 40–400 *m*/*z* range. Preliminary compound identification was performed using the NIST Mass Spectral Search Program (Version 2.2) by comparing experimental and reference mass spectra in the NIST/EPA/NIH Mass Spectral Library. Compound identification was based on matching fragmentation patterns and relative ion intensities, with only matches having a probabilistic score above 80% reported.

### 2.6. Analytical Validation

The analytical validation for matrix and analyte extension of the previously published DLLME-GC-MS method was conducted per FDA guidelines [23]. The acceptability criteria included several key components. First, the method demonstrated selectivity by providing specific retention times and identifying qualitative and quantitative ions for each analyte, ensuring accurate differentiation between compounds in the matrix. Linearity was assessed using a standard addition method with internal standard calibration within six concentration levels across a concentration range of 0.5 to 3 µg/mL for each analyte in triplicate, with strong linear relationships confirmed by high determination coefficients (r^2^). The limit of detection (LOD) was calculated as the mean of the blank signal plus 3.3 times the standard deviation of this measurement. Similarly, the limit of quantitation (LOQ) was determined as the mean of the blank signal plus 10 times the standard deviation of the blank. The method’s accuracy was evaluated at three concentration levels, with each level tested in quintuplicate, and recovery rates were calculated to assess the closeness of results to the true values. Finally, precision was evaluated by intraday and interday measurements at three concentration levels, each tested in quintuplicate, ensuring consistent results over time and under different conditions.

### 2.7. Statistical Analysis

Statistical analyses were conducted using Excel and Statistica (version 13.3) software. A box-and-whisker plot was used to present the occurrence data, showing the distribution of carbonyl compound concentrations for each sample. This type of plot displays the data’s minimum, first-quartile, median, third-quartile, and maximum values. The “box” part represents the interquartile range (IQR), which covers the middle 50% of values. The line inside the box marks the median, indicating the central value of the data set. The “whiskers” extend to the minimum and maximum values, showing the full spread of the data, while any outliers are shown as individual points outside the whiskers. This plot helps to visualize the range, central tendency, and variability in carbonyl compound concentrations across samples. A heatmap was generated to present the storage stability data, visually representing the data trends over time and across different storage conditions. The heatmap illustrated the relative stability of the analytes, with color gradients indicating the intensity of the responses. Darker shades represented higher concentrations, while lighter shades indicated lower concentrations, allowing for quick identification of significant changes in analyte levels.

## 3. Results and Discussion

### 3.1. Method Performance—Matrix and Analyte Extension

The US-DLLME-GC-MS method was previously developed in our lab to determine MDA and ACRL in beverages [22]. Herein, we studied the method’s applicability to four additional carbonyl compounds (FCHO, ACE, MGO, and DMGO) in adult formula. We used a standard adult formula as a blank sample to ensure the method’s suitability for the simultaneous extraction of these six analytes from the adult formula. First, a kinetic study was performed in triplicate to assess the formation of carbonyl-DNPH derivatives (Figure 1). MDA and ACRL reached equilibrium after 5 min of ultrasound treatment, FCHO after 10 min, while ACE, MGO, and DMGO required 20 min.

To ensure all analytes reached equilibrium and were successfully extracted as their corresponding DNPH derivatives, we set the ultrasound incubation time to 20 min. The analytical performance of the method was evaluated under these conditions (Table 1).

The method’s specificity is based on specific ions for each analyte and internal standards eluting at specific retention times previously described in this manuscript (Section 2.5). Figure 2 confirms the absence of interfering signals in the retention time region for all analytes and the internal standards. The LOD ranged from 8 to 89 ng/mL, and the LOQ ranged from 0.061 to 0.671 µg/mL. Standard addition calibrations showed excellent linearity, with determination coefficients of ≥0.9990. The method’s precision, assessed at concentrations of 0.7 µg/mL (QC1), 1.0 µg/mL (QC2), and 2.0 µg/mL (QC3), demonstrated good intraday precision with relative standard deviations (RSD) between 1.3% and 5.8%, and interday precision with RSD between 0.9% and 4.6%. Similarly, method accuracy at the same concentrations showed recoveries ranging from 98.0% to 102.6%.

Appendix A compares the analytical performance of this method with other methods for detecting carbonyl compounds in infant formula, milk, and dairy products. Single HPLC-UV determinations of MDA have been reported, using either the TBARS test or as a TBA-MDA derivative after liquid–liquid extraction or dilution. Despite being feasible and reproducible, with acceptable RSD values, these methods exhibit poorer linearity and lower recovery rates than those in this study. Similarly, single UV determinations of FCHO in milk have shown higher LOD and LOQ values than those reported here. Single MDA determination in infant formula using HPLC-MS without derivatization has been reported to have significantly lower detection limits than other HPLC methods. When paired with derivatization, HPLC-MS methods achieve even further reductions in detection limits, as reported for MGO and DA determination in baby food by Kocadağlı & Gökmen (2014), and in milk and dairy products reported by Zhang et al., 2022 [18,24]. In a previous study, we identified five carbonyl compounds, including MDA, ACRL, MGO, and DA, in infant formula using Gas-Diffusion Microextraction (GDME) with o-PDA derivatization via HPLC-UV; this method, though accurate and precise, had higher detection limits [19]. Additionally, we previously reported the development of an HS-SPME-GC-MS method for seven carbonyl and dicarbonyl compounds (including FCHO, ACE, MGO, DA, and MDA) as PFPH derivatives from infant formula, achieving detection limits similar to or slightly lower than those reported here. However, this method had higher RSD values [20].

Overall, while combining derivatization with HPLC-MS produced the lowest detection limits, the extended method reported here achieved similar or lower detection limits compared to previous GC-MS and HPLC-UV methods for milk, dairy products, and infant formula, with more accurate recoveries (closer to 100%) and comparable precision, proving the suitability of this method for simultaneous monitoring of MDA, FCHO, ACE, ACRL, MGO, and DA in adult formulas.

### 3.2. Occurrence of Carbonyl Compounds in Adult Formulas

The validated US-DLLME-GC-MS method was applied to analyze carbonyl compounds in a set of adult formula samples, with results summarized in Table 2.

All target analytes were detected, although not always at quantifiable levels; therefore, a box-and-whisker chart showing only quantifiable samples is presented for clearer comparison (Figure 3). In high-protein adult formulas, 80% of samples were quantifiable for MDA (0.53–2.89 µg/mL), FCHO (0.94–2.86 µg/mL), and ACRL (0.24–2.80 µg/mL). Additionally, 60% of samples were quantifiable for ACE (1.8–2.16 µg/mL) and MGO (0.61–1.89 µg/mL), with one sample quantifiable for DA at 2.94 µg/mL. In standard adult formulas, 50% of samples were quantifiable for MDA (1.73–2.65 µg/mL), FCHO (0.9–1.56 µg/mL), ACE (1.63–2.91 µg/mL), ACRL (1.27–1.43 µg/mL), and MGO (1.65–2.69 µg/mL), with only one sample quantifiable for DA at 0.97 µg/mL. For specialized formulations, only one enteral powder sample, intended for oral or tube administration, was quantifiable for MDA. All analytes in this sample type were quantifiable, with levels ranging from 2.0–2.09 µg/mL for MDA, 1.48–2.39 µg/mL for FCHO, 0.79–1.16 µg/mL for ACE, 1.89–2.49 µg/mL for ACRL, 1.16–2.46 µg/mL for MGO, and 1.67–2.11 µg/mL for DA.

To our knowledge, no studies have reported the occurrence levels of these compounds, specifically in adult formula. However, Kang et al. (2010) conducted short- and long-term trials to assess the effects of commercial enteral nutritional supports on the nutrition and health of stroke patients. They observed that MDA was significantly elevated in chronic stroke patients compared to healthy individuals after TBA derivatization and HPLC-UV [25].

Regarding similar foods, Bessaire et al. (2018) reported similar FCHO levels in milk powders after DNPH derivatization and HPLC-tandem mass spectrometry [26]. Custodio-Mendoza et al., (2024) observed lower levels of MDA, FCHO, and ACE in both powdered and liquid starter and follow-up infant formula after pentafluorophenyl hydrazine derivatization during HS-SPME extraction and GC-MS determination [19]. Similar ACRL, MGO, and DA levels were found in powdered starter formulas and ACRL levels in follow-up formulas after o-PDA derivatization during GDME extraction and HPLC-UV determination [20]. Similar MDA content in infant formula was reported by Pozzo et al. using TBA derivatization and HPLC-UV-FLD [27] and Cesa using the TBARS test [28]. Akıllıoğlu et al. with their microwave-assisted hydrolysis and HPLC-MS determination [15], as well as Kocadağlı et al. [24], reported comparable MGO results in the analysis of infant formulas, supporting these findings.

These findings suggest that MDA, FCHO, ACRL, ACE, MGO, and DA concentrations vary significantly across adult formula samples. The data suggest that high-protein and specialized formulations may be more susceptible to carbonyl compound formation than standard formulas. This variability highlights the importance of monitoring carbonyl compound levels in these products as potential indicators of oxidation which could affect product quality and safety.

### 3.3. Temperature-Dependent Variations of Carbonyl Compound Levels in Adult Formulas During Storage

Four samples were selected in the target stability study using the US-DLLME-GC-MS method. They included two standard adult formulas and two high-protein adult formulas from the same brand, differing only in protein content. Carbonyl compounds were detected in all samples, though quantification was only possible in HP samples at all time periods and in SAF at later stages. Results are summarized in Table 3.

In HP formulas, MDA levels increased over time under different storage conditions. At 4 °C, MDA remained relatively stable, detectable only in one HP sample, showing a slight increase at the end of 30 days. In the other HP sample, MDA became quantifiable by day 30. At room temperature, MDA accumulated more, reaching 2.01 µg/mL and 0.37 µg/mL after 30 days. Storage at 60 °C showed a similar trend, with MDA reaching comparable levels. MGO also showed accumulation over time, becoming detectable in HP samples after 7 days at 4 °C and increasing to 0.64 µg/mL and 0.98 µg/mL after 30 days. MGO levels rose further at higher temperatures, reaching 1.52–1.98 µg/mL after 30 days at room temperature and 2.54–2.78 µg/mL after 30 days at 60 °C. FCHO accumulated more gradually at 4 °C, becoming quantifiable only after 30 days at 0.5 µg/mL in one HP sample, while the other sample showed a higher range (0.94–1.93 µg/mL). FCHO accumulation intensified with higher storage temperatures, reaching up to 2.73 µg/mL at 60 °C after 30 days. In contrast, ACE and ACRL showed only slight and relatively stable increases across all temperatures.

In SAF samples, most compounds were undetectable until 7 days at both 4 °C and room temperature. MDA was below the limit of quantification in one SAF sample throughout the study, and ACRL and DA were undetectable in all SAF samples across temperatures. At 4 °C, only ACE was quantifiable in one sample after 14 days, rising to 0.37–1.38 µg/mL after 30 days. MGO was quantifiable in one SAF sample after 21 and 30 days at 0.25 and 0.71 µg/mL, respectively. In the other SAF sample, MDA, FCHO, and ACE were quantifiable only after 30 days, with concentrations of 0.28, 0.71, and 0.37 µg/mL, respectively. MDA and FCHO became quantifiable at room temperature in one SAF sample after 14 days, reaching 0.58 and 1.79 µg/mL by day 30. ACE was quantifiable in both SAF samples after 30 days, with concentrations of 0.57–1.94 µg/mL, while MGO reached 1.23 µg/mL in one sample after 21 days. At 60 °C, MDA became quantifiable after 14 days, reaching 0.32–0.88 µg/mL after 30 days. FCHO was quantifiable in one sample after 7 days, accumulating from 0.93 to 1.79 µg/mL by day 30. ACE became quantifiable in both samples after 14 and 21 days, reaching 0.62–2.27 µg/mL after 30 days, while MGO was only quantifiable in one sample after 14 days, reaching 1.43–1.94 µg/mL.

Although no data is available on the variations in carbonyl content in adult formula during one month of storage, our findings are consistent with those reported by other researchers in infant formula, milk, and dairy products. Bessaire et al. (2018) noted temperature-related differences in FCHO levels in milk powders, finding that FCHO levels were lower at room temperature than at 60 °C [26]. Cesa et al., (2015) reported a significant increase in MDA content in infant formula stored for two weeks at 55 °C [29]. Jia et al. (2018) found significant differences in content of advanced glycation end products (AGE), such as methylglyoxal and diacetyl, in milk-based infant formulas stored at 50 °C [30]. Liu and Li also observed similar variations in MGO after 60 days of storage [31]. Notably, Cheng et al. reported that even moderate increases in temperature can significantly contribute to the formation of AGEs, reaching detectable levels in infant formula milk powders, consistent with our results [32].

These findings suggest carbonyl compounds accumulate more over time in HP than in SAF, particularly at higher storage temperatures. The increase of MDA, MGO, and FCHO levels in HP formula indicates that higher protein content may promote oxidative reactions during storage, as reported in other foods [33]. Elevated temperatures further accelerate this process, with the most significant accumulation occurring at 60 °C. This implies that protein content and storage conditions (temperature and duration) are critical factors in adult formulas’ stability and potential degradation.

### 3.4. Temperature-Dependent Variations of VOC Levels in Adult Formulas During Storage

In the untargeted stability study using HS-SPME-GC-MS, identification was based on comparing the mass spectrometry (MS) patterns with those in the NIST library. The results are summarized in Appendix A, and heatmaps were created using the chromatographic peak areas obtained from a set of new SAF samples. These samples were stored for 30 days at 4 °C, room temperature, and 60 °C to facilitate comparison, as quantification was not possible (Figure 4).

A total of 40 compounds were observed, including 11 aldehydes, 10 ketones, 9 alcohols, and 2 esters, among others. Notably, hexanal, 2-heptanone, heptanal, 2-heptenal, 2-octenal, nonanal, and 2-hydroxybenzaldehyde were present in both new SAF samples. In contrast, pentyl acetate and 1-octen-3-ol were found in only one sample, while benzaldehyde, 2-pentylfuran, and octanal were found in the other sample.

Hexanal appears stable at 4 °C, with a slight increase noted in one sample stored at room temperature. However, both samples showed a significant increase in hexanal levels after 30 days at 60 °C. The other compounds in the fresh samples seem stable at both 4 °C and room temperature. In contrast, most compounds exhibited some increase when stored at 60 °C, except for nonanal and 2-hydroxybenzaldehyde, which remained stable at all temperatures. Moreover, compounds such as 1-hexanol, octanal, 1-octanol, 2-methyldecaline, and 17-octadecenoic acid appeared after 30 days of storage at all studied temperatures. Other compounds, including 1-pentanol, 1-heptanol, 2-nonanone, 1-decanol, 1-octanol, 2-decanone, decanal, γ-octalactone, 2-decenal, cyclodecanone, 6-undecanone, hexanoic acid pentyl ester, 2-undecanone, 2-undecenal, and 2-butyl-2,7-octadien-1-ol, were only present after 30 days of storage at 60 °C.

Similar VOCs, including aldehydes, ketones, alcohols, and volatile acids, were reported by Jia et al. [30] in their analysis of milk-based infant formulas, as well as by Hausner et al., (2009) in their characterization of infant formulas and breast milk [34]. Li, Zhang, and Wang also noted the formation of aldehydes and ketones in milk powders stored at various temperatures, which aligns with our findings [35]. Li et al., noticed the presence of aldehydes and ketones that influence the sensory properties of infant formulas [36]. They also found that long-term storage (up to one year) at room temperature leads to significant variations in volatile compounds, similar to our observations, with aldehydes and ketones being the most prevalent, significantly impacting the sensory attributes of infant formulas [36].

### 3.5. Risk of Exposure Assessment Based on Adult Formula Consumption

In this study, we calculated the estimated consumption of adult formula to assess potential exposure to contaminants. The analysis accounted for differences in caloric needs between males and females to avoid gender bias and individual activity levels, considering the role of adult formula as a meal replacement and the highest carbonyl content identified in the occurrence study [37]. Based on average daily caloric requirements for males and females by activity level (Appendix A) and noting that 100 mL of adult formula provides between 100–479 kcal when prepared according to the manufacturer’s instructions, we summarized the average daily intake for males weighing 70–90 kg and females weighing 55–75 kg in Table 4, expressed as µg/kg of body weight per day. We compared the average daily intakes with the tolerable or acceptable daily intakes for carbonyl compounds (see Introduction section) [38].

The risk of exposure assessment indicates that, when consuming high-protein adult formula as a meal replacement, adult women with low physical activity may experience slightly higher exposure to FCHO and DA. In contrast, for individuals with moderate to high physical activity levels, exposure to these contaminants is similar for both males and females. While exposure to most of the targeted compounds across the studied weight range and activity levels remains below the tolerable daily intake (or equivalent exposure level), both males and females would be exposed to levels of malondialdehyde twice the toxicological threshold of concern and approximately eight times the tolerable daily intake for acrolein.

## 4. Conclusions

This study highlights the need for a reliable analytical method to assess lipid peroxidation products in adult formula, a gap not addressed in prior research. By adopting a previously established method for detecting MDA and ACRL in beverages, we successfully extended it to determine six key analytes—MDA, ACRL, FCHO, ACE, MGO, and DA—in adult formulas. The method showed excellent specificity, precision, and accuracy, achieving low limits of detection and quantification.

To our knowledge, this is the first occurrence study of carbonyl compounds in adult formulas. Remarkably, high-protein formulas showed the highest detectability, with 80% of samples quantifiable for MDA, FCHO, and ACRL and 60% quantifiable for ACE and MGO. Standard formulas had a 50% quantification rate for most compounds, while only one special formulation sample, intended for oral or tube feeding, had quantifiable levels of all analytes.

The analysis of various adult formula samples revealed significant variations in the levels of carbonyl compounds when stored at different temperatures using HS-SPME-GC-MS. High-protein adult formulas consistently exhibited higher concentrations of these compounds than standard formulas. This suggests that higher protein content may enhance the formation of carbonyl compounds during storage, particularly under elevated temperatures. These findings suggest that the protein content and storage conditions—such as temperature and duration—are critical factors influencing the stability and degradation of these products.

The untargeted determination of VOCs by HS-SPME-GC-MS identified a diverse range of compounds, including various aldehydes, ketones, alcohols, and volatile acids. The results indicated that many VOCs accumulate over time, particularly in high-protein adult formulas stored at elevated temperatures. Notably, hexanal, 2-heptanone, and other compounds showed significant increases, suggesting that storage conditions critically affect the stability of these products. The presence of these VOCs, many of which are associated with sensory attributes, emphasizes the importance of understanding their impact on product quality. The findings also suggest that monitoring VOC levels can serve as an indicator of oxidation processes in adult formulas.

Our findings suggest that, once opened, adult formula should be stored in its original packaging, protected from light and moisture, and kept away from heat sources such as ovens or central heating. Formulas with higher protein content should ideally be stored in a refrigerator, particularly during the summer or in locations where the room temperature exceeds 25 °C.

Furthermore, we assessed the potential risk of exposure to these contaminants based on the estimated consumption of adult formula. Our results indicate that males and females, across all activity levels, may exceed safety thresholds for MDA and acrolein when consuming high-protein formulas. This underscores the importance of continuously monitoring carbonyl compounds in adult formulas, as high levels may compromise product quality and safety.

## Figures and Tables

**Figure 1 foods-13-03752-f001:**
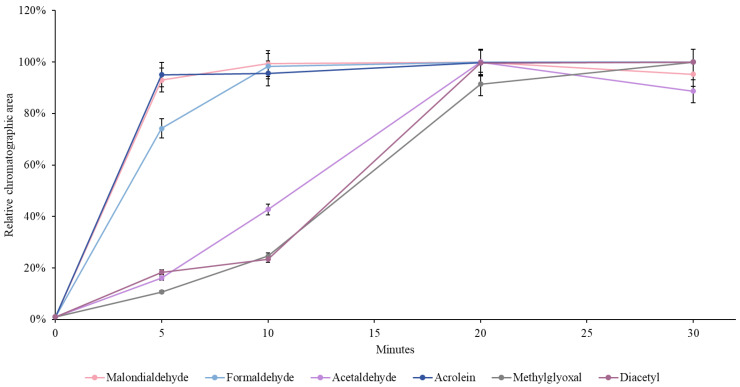
Kinetic analysis of DNPH derivatization and extraction of carbonyl compounds from adult formula using US-DLLME-GC-MS.

**Figure 2 foods-13-03752-f002:**
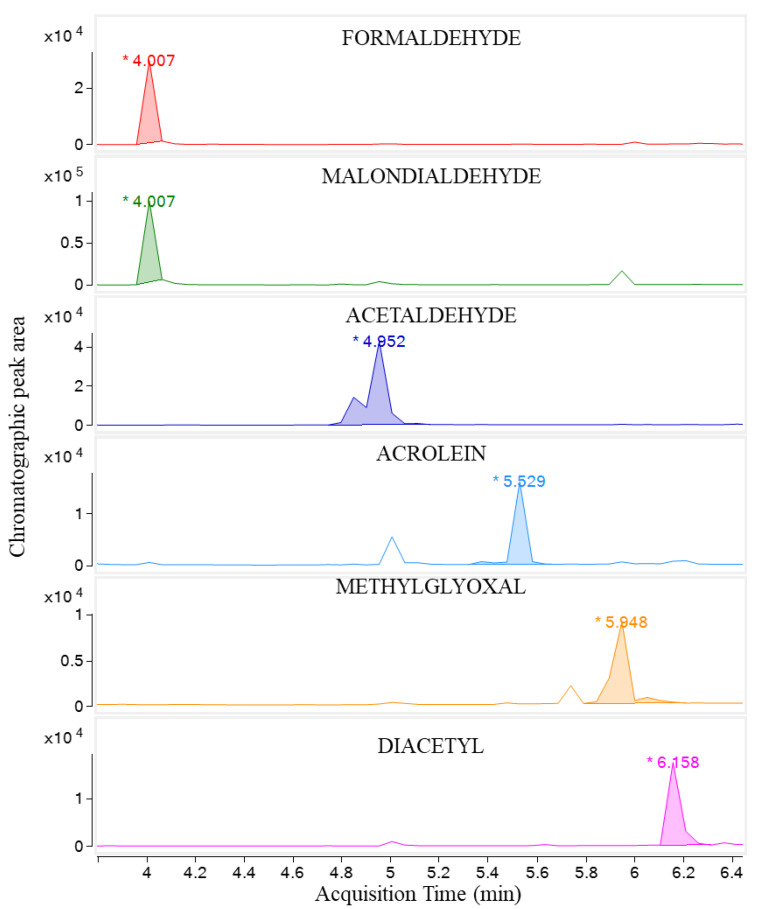
Chromatogram of standard adult formula spiked with 1 µg/mL of carbonyl compounds.

**Figure 3 foods-13-03752-f003:**
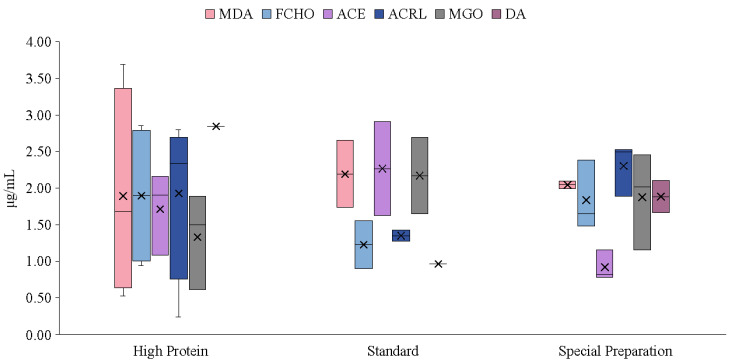
Box-and-whisker plot showing the occurrence of carbonyl compounds in adult formula samples. MDA, malondialdehyde; FCHO, formaldehyde; ACE, acetaldehyde; ACRL, acrolein; MGO, methylglyoxal; DA, diacetyl.

**Figure 4 foods-13-03752-f004:**
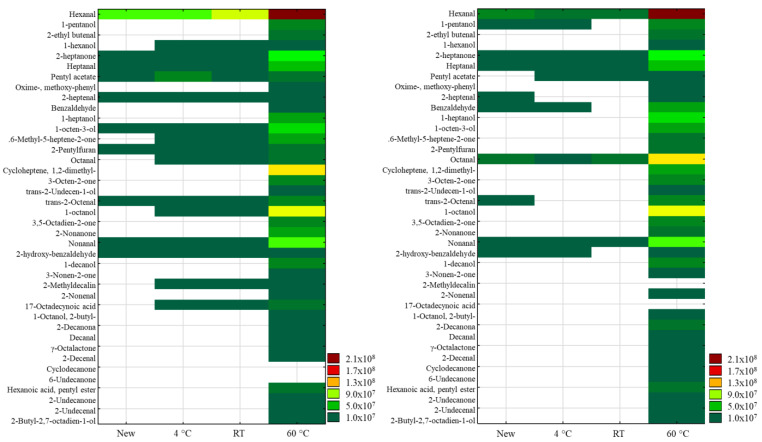
Heatmap of VOCs identified in adult formula after 1-month storage. Standard adult formula 3 (**left**), standard adult formula 4 (**right**).

**Table 1 foods-13-03752-t001:** Analytical characteristics of matrix and analyte extension study for the US-DLLME-GC-MS method.

Analyte	LOD	LOQ	Slope	Intercept	r^2^	Intraday Precision (*n* = 5) %RSD			Interday Precision (*n* = 5) %RSD			Accuracy (*n* = 5) % Recovery		
	μg/mL	μg/mL	×10^−3^			QC1	QC2	QC3	QC1	QC2	QC3	QC1	QC2	QC3
Formaldehyde	0.022	0.671	24.90	2.3749	0.9997	3.9	2.6	2.9	2.4	1.1	1.5	102.6	98.5	98.5
Malondialdehyde	0.008	0.195	2.979	0.027	0.9999	4.7	1.9	2.2	4.3	1.7	1.1	100.9	98.5	99.4
Acetaldehyde	0.089	0.340	3.968	0.0981	0.9999	2.0	5.8	0.4	4.6	0.9	1.2	98.0	99.2	100.0
Acrolein	0.017	0.061	1.999	0.0067	0.9998	5.5	4.2	2.7	3.1	3.5	1.5	100.4	99.4	98.6
Methylglyoxal	0.083	0.229	0.574	0.0499	0.9990	3.4	2.8	6.9	2.4	0.9	3.4	98.0	95.8	93.1
Diacetyl	0.065	0.184	1.552	0.0889	0.9997	1.3	2.1	1.5	1.7	2.2	1.0	97.2	97.2	98.2

LOD, limit of detection; LOQ, limit of quantification; n, number of replications; RSD, relative standard deviation; QC1, low-level quality control sample spiked at 0.7 μg/mL; QC2, mid-level quality control sample spiked at 1.0 μg/mL; QC3, high-level quality control sample spiked at 2.0 μg/mL.

**Table 2 foods-13-03752-t002:** Occurrence of carbonyl compounds in adult formula.

Adult	Malondialdehyde		Formaldehyde		Acetaldehyde		Acrolein		Methylglyoxal		Diacetyl	
Formula	μg/mL	±SD	μg/mL	±SD	μg/mL	±SD	μg/mL	±SD	μg/mL	±SD	μg/mL	±SD
HP1	2.38	0.39	2.86	0.90	2.16	0.31	2.29	0.73	1.50	0.28	ND	-
HP2	0.98	0.05	0.94	0.03	1.90	0.12	0.24	0.00	ND	-	ND	-
HP3	ND	-	ND	-	ND	-	ND	-	ND	-	ND	-
HP4	0.53	0.02	1.20	0.50	1.08	0.67	2.39	0.94	1.89	0.77	2.84	0.98
HP5	2.89	0.56	2.60	0.70	ND	-	2.80	0.17	0.61	0.08	ND	-
SAF1	2.65	0.82	0.90	0.11	1.63	0.45	1.27	0.70	1.65	0.77	ND	-
SAF2	1.73	0.54	1.56	0.31	2.91	0.23	1.43	0.35	2.69	0.82	0.97	0.27
SAF3	ND	-	ND	-	ND	-	ND	-	ND	-	ND	-
SAF4	ND	-	ND	-	ND	-	ND	-	ND	-	ND	-
SP1	2.09	0.75	1.65	0.75	1.16	0.24	2.49	0.76	2.02	0.52	2.11	0.49
SP2	2.00	0.80	1.48	0.51	0.82	0.53	1.89	0.42	2.46	0.93	1.67	0.96
SP3	ND	-	2.39	0.31	0.79	0.24	2.53	0.56	1.16	0.14	1.88	0.30

SD, standard deviation; HP, high-protein adult formula; SAF, standard adult formula; SP, special formulated adult formula; ND, non-determined as below limit of quantification.

**Table 3 foods-13-03752-t003:** Variations of carbonyl content in adult formulas during 1 month of storage at different temperatures.

	Analyte	4 °C					Room Temperature					60 °C				
		1	7	14	21	30	1	7	14	21	30	1	7	14	21	30
HP2	MDA	1.22 ± 0.41	1.19 ± 0.99	1.54 ± 0.83	1.49 ± 0.95	1.41 ± 0.40	0.86 ± 0.09	0.85 ± 0.06	1.17 ± 0.10	1.99 ± 0.04	2.01 ± 0.04	0.97 ± 0.08	1.01 ± 0.04	1.91 ± 0.06	1.91 ± 0.16	2.23 ± 0.16
	FCHO	0.95 ± 0.45	1.14 ± 0.63	1.43 ± 0.33	1.64 ± 0.13	1.93 ± 0.22	0.94 ± 0.04	1.24 ± 0.05	1.51 ± 0.03	1.93 ± 0.09	2.03 ± 0.04	0.95 ± 0.04	1.24 ± 0.04	1.74 ± 0.09	2.13 ± 0.05	2.73 ± 0.05
	ACE	1.92 ± 0.27	2.06 ± 0.03	2.08 ± 0.25	2.53 ± 0.60	2.37 ± 0.65	2.03 ± 0.23	2.07 ± 0.20	2.19 ± 0.59	2.62 ± 0.40	2.83 ± 0.03	1.96 ± 0.07	2.00 ± 0.09	2.46 ± 0.04	2.47 ± 0.16	2.94 ± 0.23
	ACRL	0.24 ± 0.53	0.24 ± 0.05	0.38 ± 0.02	0.43 ± 0.01	0.44 ± 0.01	0.26 ± 0.06	0.22 ± 0.06	0.39 ± 0.06	0.40 ± 0.03	0.45 ± 0.04	0.24 ± 0.08	0.34 ± 0.04	0.33 ± 0.06	0.42 ± 0.06	0.56 ± 0.08
	MGO	ND	0.31 ± 0.08	0.48 ± 0.05	0.57 ± 0.02	0.64 ± 0.03	ND	0.32 ± 0.04	1.22 ± 0.03	1.75 ± 0.06	1.98 ± 0.01	0.23 ± 0.05	1.49 ± 0.01	1.60 ± 0.03	2.08 ± 0.09	2.78 ± 0.06
	DA	ND	ND	ND	ND	ND	ND	ND	ND	ND	ND	ND	ND	ND	ND	ND
HP3	MDA	ND	ND	ND	ND	0.28 ± 0.04	ND	ND	ND	ND	0.37 ± 0.03	ND	ND	0.30 ± 0.04	0.30 ± 0.06	0.40 ± 0.02
	FCHO	ND	ND	ND	ND	0.50 ± 0.08	0.77 ± 0.01	0.76 ± 0.02	0.80 ± 0.04	0.83 ± 0.08	0.92 ± 0.01	0.84 ± 0.07	0.88 ± 0.09	0.92 ± 0.09	0.96 ± 0.05	0.97 ± 0.07
	ACE	0.99 ± 0.01	1.44 ± 0.25	1.32 ± 0.27	1.36 ± 0.15	1.38 ± 0.17	1.13 ± 0.06	1.51 ± 0.12	1.81 ± 0.10	1.95 ± 0.33	2.39 ± 0.25	1.03 ± 0.31	1.65 ± 0.37	1.91 ± 0.28	2.15 ± 0.25	2.57 ± 0.17
	ACRL	ND	ND	ND	ND	ND	ND	ND	ND	ND	ND	ND	ND	ND	ND	ND
	MGO	ND	0.35 ± 0.02	0.67 ± 0.17	0.71 ± 0.15	0.98 ± 0.10	0.65 ± 0.03	1.05 ± 0.25	1.19 ± 0.13	1.23 ± 0.44	1.52 ± 0.27	0.82 ± 0.04	1.01 ± 0.04	1.43 ± 0.05	2.34 ± 0.09	2.54 ± 0.07
	DA	ND	ND	ND	ND	ND	ND	ND	ND	ND	ND	ND	ND	ND	ND	ND
SAF3	MDA	ND	ND	ND	ND	0.28 ± 0.06	ND	ND	0.28 ± 0.04	0.48 ± 0.05	0.58 ± 0.03	ND	ND	0.36 ± 0.01	0.57 ± 0.03	0.88 ± 0.09
	FCHO	ND	ND	ND	ND	0.71 ± 0.02	ND	ND	1.26 ± 0.01	0.13 ± 0.01	1.79 ± 0.09	ND	0.93 ± 0.09	1.41 ± 0.05	1.55 ± 0.05	1.79 ± 0.02
	ACE	ND	ND	ND	ND	0.37 ± 0.08	ND	ND	ND	ND	0.57 ± 0.08	ND	ND	ND	0.55 ± 0.05	0.62 ± 0.01
	ACRL	ND	ND	ND	ND	ND	ND	ND	ND	ND	ND	ND	ND	ND	ND	ND
	MGO	ND	ND	ND	ND	ND	ND	ND	ND	ND	ND	ND	ND	ND	ND	ND
	DA	ND	ND	ND	ND	ND	ND	ND	ND	ND	ND	ND	ND	ND	ND	ND
SAF4	MDA	ND	ND	ND	ND	ND	ND	ND	ND	ND	ND	ND	ND	ND	ND	0.32 ± 0.02
	FCHO	ND	ND	ND	ND	ND	ND	ND	ND	ND	ND	ND	ND	ND	ND	0.62 ± 0.02
	ACE	ND	ND	1.12 ± 0.01	1.26 ± 0.07	1.38 ± 0.06	ND	ND	1.81 ± 0.23	1.85 ± 0.27	1.94 ± 0.44	ND	ND	1.91 ± 0.09	2.15 ± 0.27	2.27 ± 0.57
	ACRL	ND	ND	ND	ND	ND	ND	ND	ND	ND	ND	ND	ND	ND	ND	ND
	MGO	ND	ND	ND	0.25 ± 0.01	0.71 ± 0.08	ND	ND	ND	0.88 ± 0.02	1.23 ± 0.05	ND	ND	1.43 ± 0.25	1.54 ± 0.43	1.94 ± 0.13
	DA	ND	ND	ND	ND	ND	ND	ND	ND	ND	ND	ND	ND	ND	ND	ND

HP, high-protein adult formula; SAF, standard adult formula; MDA, malondialdehyde; FCHO, formaldehyde; ACE, acetaldehyde; ACRL, acrolein; MGO, methylglyoxal; DA, diacetyl; ND, non-determined as it was below the limit of quantification.

**Table 4 foods-13-03752-t004:** Average daily intake of carbonyl compounds for males and females by activity level.

Level of Physical Activity	Malondialdehyde		Formaldehyde		Acetaldehyde		Acrolein		Methylglyoxal ^a^		Diacetyl	
	μg/Kg bw/day		μg/Kg bw/day		μg/Kg bw/day		μg/Kg bw/day		μg/Kg bw/day		μg/Kg bw/day	
	M	F	M	F	M	F	M	F	M	F	M	F
Low	53.5 *	55.3 *	52.9	69.8	53.8	55.7	51.8 *	53.6 *	49.8	51.5	52.5	67.7
Moderate	58.9 *	61.4 *	58.3	60.8	59.3	61.8	57.1 *	59.5 *	54.9	57.1	57.9	60.3
High	67.9 *	67.5 *	67.2	66.8	68.3	67.9	65.7 *	65.4 *	63.2	62.8	66.7	66.3

bw, body weight; M, male; F, female; * above tolerable daily intake (or analog exposure level); ^a^ compared to glyoxal equivalent due to lack of exposure level.

## Data Availability

The data presented in this study are available on request from the corresponding author due to privacy.

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
