# Peer review of "Assessment of Lipid Peroxidation Products in Adult Formulas: GC-MS Determination of Carbonyl and Volatile Compounds Under Different Storage Conditions"

_foods, 2024, doi:10.3390/foods13233752_

Round 1
Reviewer 1 Report
Comments and Suggestions for Authors
The authors investigated carbonyl compounds and volatile organic compounds (VOCs) to ensure safety and quality. The analysis was performed using gas chromatography-mass spectrometry on samples stored at various temperatures over a month. Ultrasound-assisted dispersive liquid-liquid microextraction was employed for analyzing carbonyl compounds, while headspace solid-phase microextraction was used for untargeted VOC analysis. Consistent with FDA guidelines, the methods demonstrated excellent detection limits, quantitation, linearity, accuracy, and precision. The results revealed higher concentrations of carbonyl compounds in high-protein formulas and identified malondialdehyde, acrolein, and formaldehyde in 80% of the samples.
The manuscript is well written, the experiments are well planed and presented. Data and its discussion are convincing. I recommend publishing after minor revisions.
Note regarding literature - please remove the introductory text.
In the description of Table 1, Figure 1, Figure 2, Figure 3 and Figure 4 - please change the text - to normal in the sentence.
Please expand the description of Table 4.
Author Response
We sincerely thank the reviewer for thoroughly evaluating our manuscript and appreciate the positive feedback. All suggestions have been carefully addressed, and the corresponding modifications have been made in the text, with specific line numbers provided to facilitate the review process.
See attachment.

Reviewer 2 Report
Comments and Suggestions for Authors
This paper investigated ‘Assessment of Lipid Peroxidation Products in Adult Formulas: GC-MS Determination of Carbonyl and Volatile Compounds Under Different Storage Conditions’ is interesting. The comments are as follows:
1) Please give some suggestions about the storage condition.
2) Please add the structural formula of these contaminants.
3) Please briefly explore the formation mechanism of pollutants.
Author Response
Thank you for your valuable feedback. We have incorporated all your comments into the text and included specific line numbers to assist with the review process.
See attachment.

Reviewer 3 Report
Comments and Suggestions for Authors
This manuscript reports the determination of lipid peroxidation products in adult formulas by GC-MS. This work is clearly novel, significant, and within the aims and scope of Foods. The novelty of this work focuses upon the evaluation of the carbonyl compounds in various storage conditions and the conclusion that elevated temperature and high protein content enhance the production of toxic chemicals. The authors are well qualified to perform this study and the work was carefully validated. The quality of the English is excellent. The tables and figures are of publication quality and essential for the presentation of the results. The references are appropriate and relevant to this study. This work has considerable significance for human health, as the consumers of these products generally have health conditions, and hence an understanding of these toxic chemicals is essential.
Author Response
We thank the reviewer for their thorough review and appreciate their positive feedback.